# PPanGGOLiN: Depicting microbial diversity via a partitioned pangenome graph

**Guillaume Gautreau**[1], **Adelme Bazin**[1], **Mathieu Gachet**[1], **Rémi Planel**[1¤a],
**Laura Burlot**[1], **Mathieu Dubois**[1], **Amandine Perrin**[2,3], **Claudine Médigue**[1],
**Alexandra Calteau**[1], **Stéphane Cruveiller**[1¤b], **Catherine Matias**[4], **Christophe Ambroise**[5],
**Eduardo P. C. Rocha**[2], **David Vallenet**[1]*

**1** LABGeM, Génomique Métabolique, CEA, Genoscope, Institut François Jacob, Université d'Évry, Université
Paris-Saclay, CNRS, Evry, France, **2** Microbial Evolutionary Genomics, Institut Pasteur, CNRS, UMR3525,
Paris, France, **3** Sorbonne Université, Collège doctoral, Paris, France, **4** Laboratoire de Probabilités,
Statistique et Modélisation, Sorbonne Université, Université de Paris, Centre National de la Recherche
Scientifique, Paris, France, **5** Laboratoire de Mathématiques et Modélisation d'Evry, UMR CNRS 8071,
Université d'Evry Val d'Essonne, Evry, France

¤a Current address: Hub de Bioinformatique et Biostatistique - Département Biologie Computationnelle,
Institut Pasteur, USR 3756 CNRS, Paris, France
¤b Current address: PathoQuest SAS, BioPark – bâtiment B, 11 rue Watt, 75013 Paris, France
* vallenet@genoscope.cns.fr

pcbi.1007732

**Data Availability Statement:** Archaeal and
bacterial genomes were downloaded from the
NCBI FTP server (ftp://ftp.ncbi.nlm.nih.gov/
genomes/genbank) 17 April 2019. Metagenome-
Assembled Genomes were downloaded from

## Abstract

The use of comparative genomics for functional, evolutionary, and epidemiological studies
requires methods to classify gene families in terms of occurrence in a given species. These
methods usually lack multivariate statistical models to infer the partitions and the optimal
number of classes and don't account for genome organization. We introduce a graph struc-
ture to model pangenomes in which nodes represent gene families and edges represent
genomic neighborhood. Our method, named PPanGGOLiN, partitions nodes using an
Expectation-Maximization algorithm based on multivariate Bernoulli Mixture Model coupled
with a Markov Random Field. This approach takes into account the topology of the graph
and the presence/absence of genes in pangenomes to classify gene families into persistent,
cloud, and one or several shell partitions. By analyzing the partitioned pangenome graphs of
isolate genomes from 439 species and metagenome-assembled genomes from 78 species,
we demonstrate that our method is effective in estimating the persistent genome. Interest-
ingly, it shows that the shell genome is a key element to understand genome dynamics, pre-
sumably because it reflects how genes present at intermediate frequencies drive adaptation
of species, and its proportion in genomes is independent of genome size. The graph-based
approach proposed by PPanGGOLiN is useful to depict the overall genomic diversity of
thousands of strains in a compact structure and provides an effective basis for very large
scale comparative genomics. The software is freely available at https://github.com/labgem/
PPanGGOLiN.

https://opendata.lifebit.ai/table/SGB. All analyses described here were run using PPanGGOLiN software (version 1.0). PPanGGOLiN source code is freely available from https://github.com/labgem/PPanGGOLiN under a CeCILL license. All relevant data are within the manuscript and its Supporting Information files.

**Funding:** This research was supported in part by the IRTELIS and Phare PhD programs of the French Alternative Energies and Atomic Energy Commission (CEA) for GG and AB respectively, the French Government "Investissements d'Avenir" programs (namely FRANCE GENOMIQUE [ANR-10-INBS-09-08], the INSTITUT FRANÇAIS DE BIOINFORMATIQUE [ANR-11-INBS-0013], and the Agence Nationale de la Recherche [Projet ANR-16-CE12-29 for EPCR]). The funders had no role in study design, data collection and analysis, decision to publish, or preparation of the manuscript.

**Competing interests:** The authors have declared that no competing interests exist.

## Author summary

Microorganisms have the greatest biodiversity and evolutionary history on earth. At the genomic level, it is reflected by a highly variable gene content even among organisms from the same species which explains the ability of microbes to be pathogenic or to grow in specific environments. We developed a new method called PPanGGOLiN which accurately represents the genomic diversity of a species (i.e. its pangenome) using a compact graph structure. Based on this pangenome graph, we classify genes by a statistical method according to their occurrence in the genomes. This method allowed us to build pangenomes even for uncultivated species at an unprecedented scale. We applied our method on all available genomes in databanks in order to depict the overall diversity of hundreds of species. Overall, our work enables microbiologists to explore and visualize pangenomes alike a subway map.

## Introduction

The analyses of the gene repertoire diversity of species—their pangenome—have many applications in functional, evolutionary, and epidemiological studies [1, 2]. The core genome is defined as the set of genes shared by all the genomes of a taxonomic unit (generally a species) whereas the accessory (or variable) genome contains genes that are only present in some genomes. The latter is crucial to understand bacterial adaptation as it contains a large repertoire of genes that may confer distinct traits and explain many of the phenotypic differences across species. Most of these genes are acquired by horizontal gene transfer (HGT) [3]. This usual dichotomy between core and accessory genomes does not consider the diverse ranges of gene frequencies in a pangenome. The main problem in using a strict definition of the core genome is that its size decreases as more genomes are added to the analysis [4] due to gene loss events and technical artifacts (i.e. sequencing, assembly or annotation issues). As a consequence, it was proposed in the field of synthetic biology to focus on persistent genes, i.e. those conserved in a large majority of genomes [5]. The persistent genome is also called the soft core [6], the extended core [7, 8] or the stabilome [9]. These definitions advocate for the use of a threshold frequency of a gene family within a species above which it is considered as *de facto* core gene. Persistent gene families are usually defined as those present in a range comprised between 90% [10] and 99% [11] of the strains in the species. This approach addresses some problems of the original definition of core genome but requires the setting of an appropriate threshold. The gene frequency distribution in pangenomes is extensively documented [7, 8, 12–16]. Due to the variation in the rates of gene loss and gain of genes, the gene frequencies tend to show an asymmetric U-shaped distribution regardless of the phylogenetic level and the clade considered (with the exception of few species having non-homogeneous distributions as described in [17]). Thereby, as proposed by Koonin and Wolf [12] and formally modeled by Collins and Higgs [14], the pangenome can be split into 3 classes: (1) persistent genome, for the gene families present in almost all genomes; (2) shell genome, for gene families present at intermediate frequencies in the species; (3) cloud genome, for gene families present at low frequency in the species.

The study of pangenomes in microbiology now relies on the comparison of hundreds to thousands of genomes of a single species. The analysis of this massive amount of data raises computational and algorithmic challenges that can be tackled because genomes within a species have many homologous genes and it is possible to design new compact ways of representing and manipulating this information. As suggested by Chan *et al.* [18], a consensus representation of multiple genomes would provide a better analytical framework than using individual

reference genomes. Among others, this proposition has led to a paradigm shift from the usual linear representation of reference genomes to a representation as variation graphs (also named "genome graphs" or "pangenome graphs") bringing together all the different known variations as multiple alternative paths. Methods [19–21] have been developed aiming at factorizing pangenomes at the genome sequence-level to capture all the nucleotide variations in a graph that enables variant calling and improves the sensitivity of the read mapping (summarized in [22]).

The method presented here, named PPanGGOLiN (Partitioned PanGenome Graph Of Linked Neighbors), introduces a new representation of the gene repertoire variation as a graph, where each node represents a family of homologous genes and each edge indicates a relation of genetic contiguity. PPanGGOLiN fills the gap between the standard pangenomic approach (that uses a set of independent and isolated gene families) and sequence-level pangenome graph (as reviewed in [23]). The interest of a gene-level graph compared to a sequence graph is that it provides a much more compact structure in clades where gene gains and losses are the major drivers of adaptation. This comes at the cost of disregarding polymorphism in genes and ignoring variation in intergenic regions and introns. However, the genomes of prokaryotes have very small intergenic regions and are almost devoid of introns justifying a focus on the variation of gene repertoires [12], which can be complemented by analysis of intergenic and intragenic polymorphism. PPanGGOLiN uses a new statistical model to classify gene families into persistent, cloud, and one or several shell partitions. To the best of our knowledge three statistical methods are available to partition a pangenome. Two of them use probabilistic models that partition dichotomously the pangenome only into core and accessory components [24, 25]. Conversely, the method proposed and implemented by Snipen *et al.* [26, 27] (micropan R package) classifies a pangenome in *K* partitions using a Binomial Mixture Model relying on gene family frequencies. Unlike these three methods, PPanGGOLiN is not based on frequencies but combines both the patterns of occurrence of gene families and the pangenome graph topology to perform the classification. In the following sections we present an overview of the method, an illustration of a pangenome graph and then the partitioning of a large set of prokaryotic species from GenBank. We evaluate the relevance of the persistent genome computed by PPanGGOLiN in comparison to the classical soft core genome. Next, we illustrate the importance of the shell structure and dynamics in the study of the evolution of microbial genomes. Finally, we compare GenBank results to the ones obtained with Metagenome-Assembled Genomes (MAGs) to validate the use of PPanGGOLiN for metagenomic applications.

## Results and discussion

### Overview of the PPanGGOLiN method

PPanGGOLiN builds pangenomes for large sets of prokaryotic genomes (i.e. several thousands) through a graphical model and a statistical method to classify gene families into three classes: persistent, cloud, and one or several shell partitions. It uses as input a set of annotated genomes with their coding regions classified in homologous gene families. As depicted in Fig 1, PPanGGOLiN integrates information on protein-coding genes and their genomic neighborhood to build a graph where each node is a gene family and each edge is a relation of genetic contiguity (two families are linked in the graph if they contain genes that are neighbors in the genomes). Thanks to this graphical model, the structure of the pangenome is resilient to fragmented assemblies: an assembly gap in one genome can be offset by information from other genomes, thus maintaining the link in the graph. To partition this graph, we established a statistical model taking into consideration that persistent genes share conserved genomic organizations along genomes (i.e. synteny conservation) [28] and that horizontally transferred genes

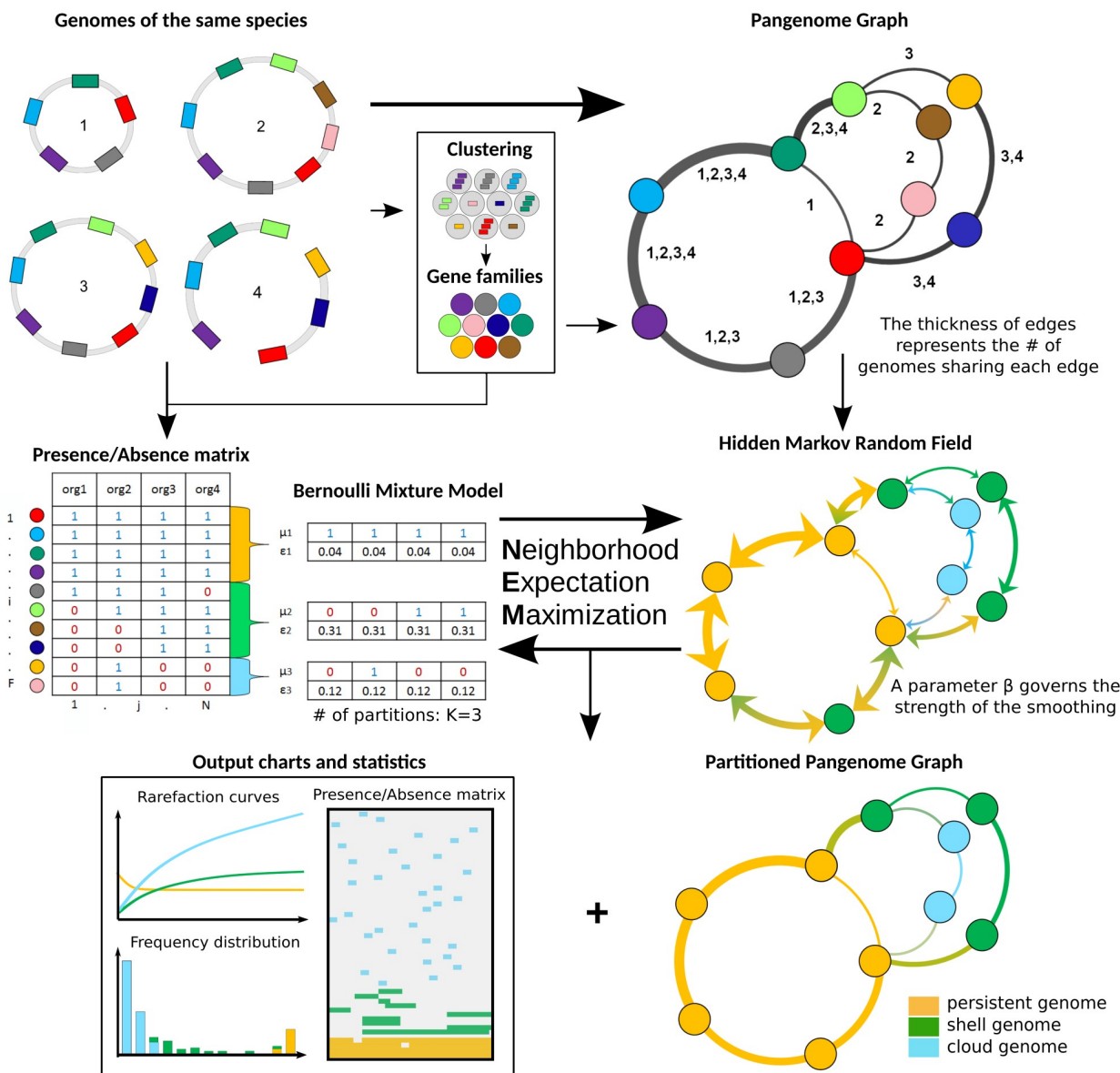

**Fig 1. Flowchart of PPanGGOLiN on a toy example of 4 genomes.** The method requires annotated genomes of the same species with their genes clustered into homologous gene families. Annotations and gene families can be predicted by PPanGGOLiN or directly provided by the user. Based on these inputs, a pangenome graph is built by merging homologous genes and their genomic links. Nodes represent gene families and edges represent genomic neighborhood. The edges are labeled by identifiers of genomes sharing the same gene neighborhood. In parallel, gene families are encoded as a presence/absence matrix that indicates for each family whether or not it is present in the genomes. The pangenome is then divided into *K* partitions (*K* = 3 in this example) by estimating the best partitioning parameters through an Expectation-Maximization algorithm. The method involves the maximization of the likelihood of a multivariate Bernoulli Mixture Model taking into account the constraint of a Markov Random Field (MRF). The MRF network is given by the pangenome graph and it favors two neighbors to be more likely classified in the same partition. At the end of this iterative process, PPanGGOLiN returns a partitioned pangenome graph where persistent, shell and cloud partitions are overlaid on the neighborhood graph. In addition, many tables, charts and statistics are provided by the software. The number of partitions (*K*) can either be provided by the user or determined by the algorithm.

(i.e. shell and cloud genes) tend to insert preferentially in a few chromosomal regions (hotspots) [29]. Thereby, PPanGGOLiN favors two gene families that are consistent neighbors in the graph to be more likely classified in the same partition. This is achieved by a hidden Markov Random Field (MRF) whose network is given by the pangenome graph. In parallel, the

pangenome is also represented as a binary Presence/Absence (P/A) matrix where the rows correspond to gene families and the columns to genomes. Values are 1 for the presence of at least one member of the gene family and 0 otherwise. This P/A matrix is modeled by a multivariate Bernoulli Mixture Model (BMM). Its parameters are estimated via an Expectation-Maximization (EM) algorithm taking into account the constraints imposed by the MRF. Each gene family is then associated to its closest partition according to the BMM. This results in a partitioned pangenome graph made of nodes that are classified as either persistent, shell or cloud. The strength of the MRF constraints increases according to a parameter called $\beta$ (if $\beta = 0$, the effect of the MRF is disabled and the partitioning only relies on the P/A matrix) and it depends on the weight of the edges of the pangenome graph which represents the number of gene pairs sharing the neighborhood. Another originality of our method is that, even if the number of partitions ($K$) is estimated to be equal to 3 (persistent, shell, cloud) in most cases (see 'Analyses of the most represented species in databanks' section), more partitions can be used if the pangenome matrix contains several contrasted patterns of P/A. These additional partitions are considered to belong to the shell genome and reflect a heterogeneous structure of the shell (see 'Shell structure and dynamics' section).

## Illustration of a partitioned pangenome graph depicting the *Acinetobacter baumannii* species

We computed the pangenome of 3 117 *Acinetobacter baumannii* genomes from GenBank using PPanGGOLiN. For the persistent, shell and cloud genomes, we obtained 3 084, 1 529 and 64 833 gene families, respectively. If we compare our results with those of Chan *et al.* study [18], the size of the persistent genome predicted by PPanGGOLiN is included in their soft core estimation ranging from 2 833 (95% of presence) to 3 126 (75% of presence) gene families using 249 *A. baumannii* genomes. On the partitioned pangenome graph built with PPanGGOLiN (Fig 2), the gene families classified as persistent (orange nodes) correspond to the conserved paths that are interrupted by many islands composed of shell (green nodes) and cloud genomes (blue nodes). These islands appear to be frequently inserted in hotspots of the persistent genome thus pinpointing regions of genome plasticity. The average node degree within the same partition is 2.80 for the persistent genome while the shell genome has a higher average degree (3.95, P = 5.0e-6 with a bilateral unpaired 2-sample Student's t test) and the cloud a lower one (1.97, P = 3.3e-40 with the same test). The shell genome is the most diversified in terms of network topology with many interconnections between families reflecting a mosaic composition of regions from different HGT events [29]. The major part of the cloud has a shell-like graph topology with a large connected component containing 60% of the nodes. In addition, the cloud also contains isolated components that are nearly linear (3 606 components having on average 4.25 nodes) and singletons (10 575 nodes), presumably because it includes very recently acquired genetic material. Finally, large families of mobile genes, mostly transposable elements, can be easily detected because they constitute hubs (i.e. highly connected nodes) in the graph. They vary rapidly their genetic neighborhoods and can be found in multiple loci.

As an example of a more detailed analysis that can be done using the graph, a zoom on a region containing the genes required for the synthesis of capsular polysaccharides is highlighted in Fig 2. *A. baumannii* strains are involved in numerous nosocomial infections and their capsule plays key roles in the overall fitness and pathogenicity. Indeed, it protects the bacteria against environmental stresses, host immune responses and can confer resistance to some antimicrobial compounds [30]. Over one hundred distinct capsule types and their corresponding genomic organization have been reported in *A. baumannii* [31]. A zoom on this

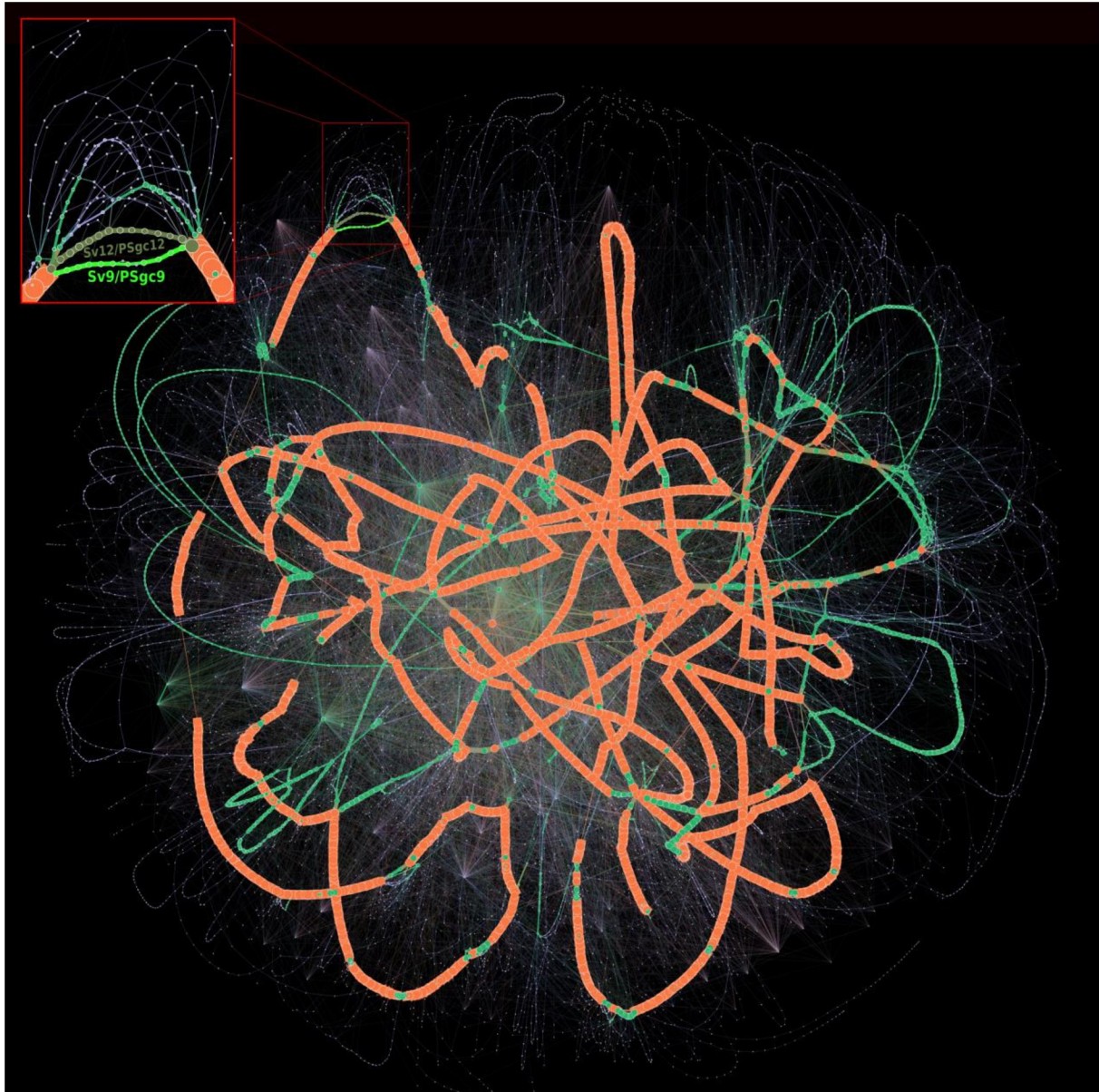

**Fig 2. Partitioned pangenome graph of 3 117 *Acinetobacter baumannii* genomes.** This partitioned pangenome graph of PPanGGOLiN displays the overall genomic diversity of 3 117 *Acinetobacter baumannii* strains from GenBank. Edges correspond to genomic colocalization and nodes correspond to gene families. The thickness of the edges is proportional to the number of genomes sharing that link. The size of the nodes is proportional to the total number of genes in each family. The edges between persistent, shell and cloud nodes are colored in orange, green and blue, respectively. Nodes are colored in the same way. The edges between gene families belonging to different partitions are shown in mixed colors. For visualization purposes, gene families with less than 20 genes are not shown on this figure although they comprise 84.68% of the nodes (families mostly composed of a single gene). The frame in the upper left corner shows a zoom on a branching region where multiple alternative shell and cloud paths are present in the species. This region is involved in the synthesis of the major polysaccharide antigen of *A. baumannii*. The two most frequent paths (Sv12/PSgc12 and Sv9/PSgc9) are highlighted in khaki and fluo green. The Gephi software ([https:// gephi.org](https://gephi.org)) [32] with the ForceAtlas2 algorithm [33] was used to compute the graph layout with the following parameters: Scaling = 8000, Stronger Gravity = True, Gravity = 4.0, Edge Weight influence = 1.3.

region of the graph shows a wide variety of combinations of genes for the synthesis of capsular polysaccharides. Based on the 3 117 *A. baumannii* genomes available in GenBank, we detected 229 different paths, sharing many common portions, but only a few are conserved in the species (only 24 paths are covered by more than 10 genomes). Among them, two alternative shell

paths seem to be particularly conserved (from the *gnaA* to the *weeH* genes in the figure 3 of [31]). Based on the nomenclature of [31], one (colored in khaki green in the Fig 2) corresponds to the serovar called PSgc12, contains 14 gene families of the shell genome and is fully conserved in 581 genomes. The other (colored in fluo green in the Fig 2) corresponds to the serovar PSgc9 (equivalent to PSgc7), contains 11 gene families of the shell genome and is fully conserved in 408 genomes. This analysis illustrates how the partitioned pangenome graph of PPanGGOLiN can be useful to study the plasticity of genomic regions. Thanks to its compact structure in which genes are grouped into families while preserving their genomic neighborhood information, it summarizes the diversity of thousands of genomes in a single picture and allows effective exploration of the different paths among regions or genes of interest.

## Analyses of the most represented species in databanks

We used PPanGGOLiN to analyze all prokaryotic species of GenBank for which at least 15 genomes were available. This is the minimal number of genomes we recommend to ensure a relevant partitioning. The quality of the genomes was evaluated before their integration in the graph to avoid taxonomic assignation errors and contamination that can have a major impact on the analysis of pangenomes (see Materials and methods). This resulted in a dataset of 439 species pangenomes, whose metrics are available in S1 File. We focused our analysis on the 88 species containing at least 100 genomes (Fig 3). This data was used for in-depth analysis of persistent and shell genomes (see the two next sections). Proteobacteria, Firmicutes and Actinobacteria are the most represented phyla in this dataset and comprise a variety of species, genome sizes and environments. In contrast, Spirochaetes, Bacteroidetes and Chlamydiae phyla are represented by only one or two species (*Leptospira interrogans*, *Bacteroides fragilis*, *Flavobacterium psychrophilum* and *Chlamydia trachomatis*). For each species, we computed the median and interquartile range of persistent, shell and cloud families in the genomes. As expected, we observed a large variation in the range of these values: from pathogens with reduced genomes such as *Bordetella pertussis* or *C. trachomatis* which contain only a small fraction of variable gene families (less than ≈5% of shell and cloud genomes) to commensal or environmental bacteria such as *Bifidobacterium longum* and *Burkholderia cenocepacia* whose shell represents more than ≈35% of the genome. Furthermore, for a few species the number of estimated partitions (*K*) is greater than 3 (11 out of 88 species), especially for those with a higher fraction of shell genome. Hence, our method provides a statistical justification for the use of three partitions as a default in pangenome analyses, while indicating that species with large shell content might be best modeled using more partitions (see 'Shell structure and dynamics' section).

## Estimation of the persistent genome in comparison to the soft core approach

To demonstrate the added value of PPanGGOLiN, we compared our statistical method to a classical approach where persistent genes are those present in at least 95% of the genomes (generally called the soft core approach). Indeed, this threshold is very often used in pangenomic studies probably because it is the default parameter in Roary [34] which is to date the most cited software to build bacterial pangenomes. In the 88 studied species, the number of persistent gene families is greater than or equal to the soft core with an average of 11% (SD = 9%) of additional families (see Fig 3 and S1 File). Furthermore, persistent gene families include those of the soft core with the exception of very few gene families (12 families in total for all studied species). The gene family frequencies in each of the 88 pangenomes are available in S1 Fig. For four species, *Pseudomonas stutzeri*, *Clostridium perfringens*, *Clostridium*

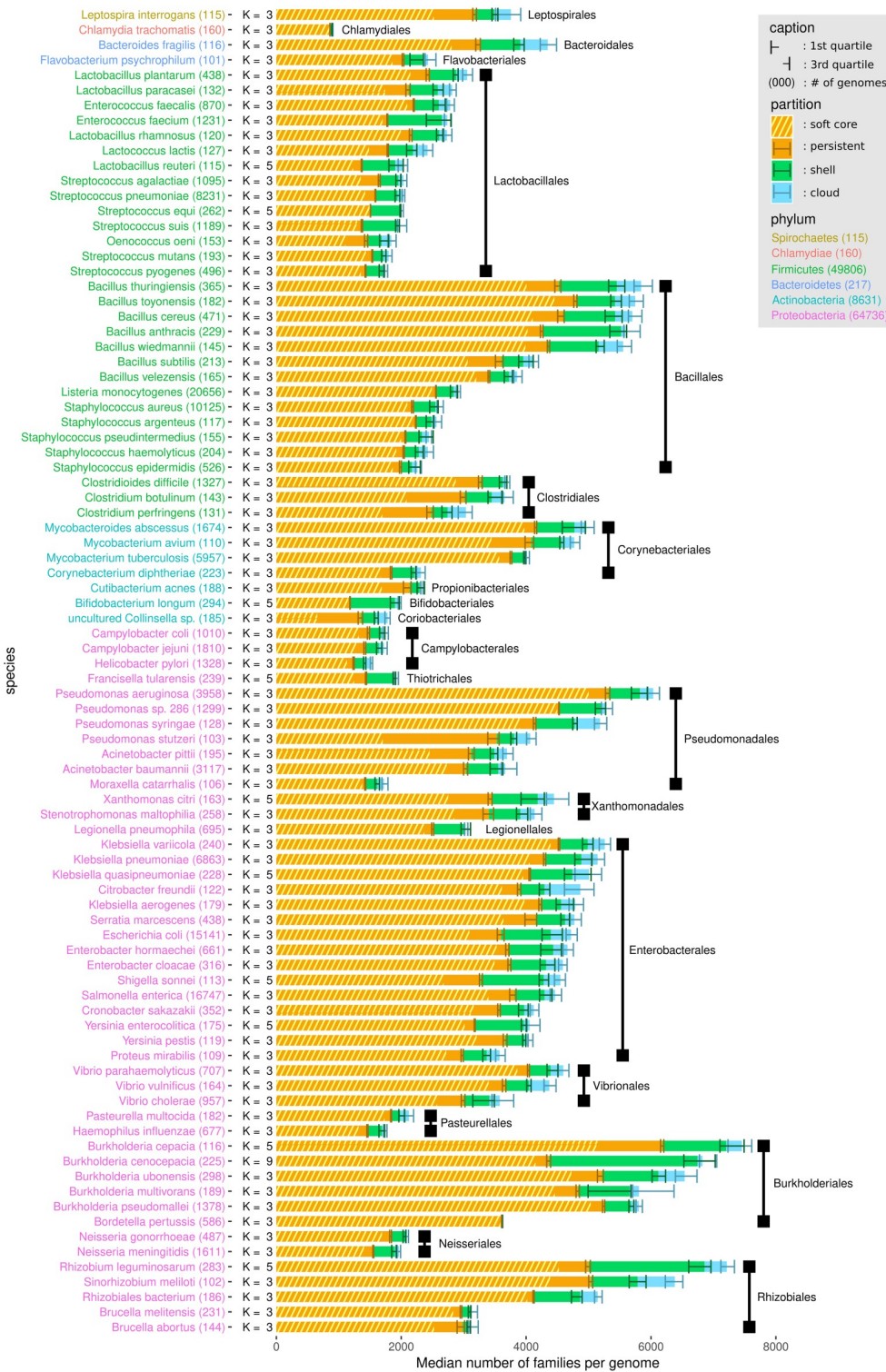

**Fig 3. Distribution of PPanGGOLiN partitions in the genomes of the most represented species in GenBank.** Each horizontal bar shows the median number of gene families per genome among the different PPanGGOLiN partitions (persistent, shell and cloud) in the 88 most represented species in GenBank (having at least 100 genomes). The error bars represent the interquartile ranges. Hatched areas on the persistent genome bars show the median number of gene families for the soft core (≥95% of presence). The species names are colored according to their phylum and sorted by taxonomic order and then by decreasing cumulative bar size. Next to the species names, the number of genomes is indicated in brackets and the number of partitions (*K*) that was automatically determined by PPanGGOLiN is also shown.

*botulinum* and *Colinsella sp.*, the size of the soft core genome is unexpectedly small and represents less than 55% of the genomes whereas it is above 75% for the PPanGGOLiN persistent. For the first three species, this could be due to sampling effects and species heterogeneity. For the last one (*Colinsella sp.*), this could be explained by the fact that the species is made of incomplete genomes from metagenomes (i.e. MAGs) that were submitted as complete genomes in GenBank.

For an in-depth comparison of these approaches, we performed multiple resamplings of the genome dataset for each species in order to measure the variability of the pangenome metrics and the impact of genome sampling according to an increasing number of genomes considered in the analyses (hereafter called rarefaction curves, see Materials and methods for more details and S2 Fig as an example for *Lactobacillus plantarum*). These rarefaction curves indicate whether the number of families tends to stabilize, increase or decrease. To this end, the curves were fit with the Heaps' law where $\gamma$ represents the growth tendency [35] (hereafter called $\gamma$-tendency). The persistent component of a pangenome is supposed to stabilize after the inclusion of a certain number of genomes, which means it has a $\gamma$-tendency close to 0. In addition, interquartile range (IQR) areas along the rarefaction curves were computed to estimate the variability of the predictions in relation to the sampling. Small IQR areas mean that the predictions are stable and resilient to sampling. Using these metrics, the PPanGGOLiN predictions of the persistent genome were evaluated in comparison to the soft core approach.

We observed that the $\gamma$-tendency of the PPanGGOLiN persistent is closer to 0 than that of the soft core approach (mean of absolute $\gamma$-tendency = 9.1e-3 versus 2.5e-2, P = 1.5e-9 with a one-sided paired 2-sample Student's t-test) with a lower standard deviation error too (mean = 5.3e-04 versus 2.1e-03, P = 9.5e-11 with one-sided paired 2-sample Student's t-test) (see Fig 4 and S1 File). A major problem of the soft core approach is that the $\gamma$-tendency is high for many species (32 species have a $\gamma$-tendency above 0.025), suggesting that the size of the persistent genome is not stabilized and tends to be underestimated. Besides, the IQR area of the PPanGGOLiN prediction is far below the one of the soft core genome (mean = 4906.6 versus 11645.9, P = 8.9e-07 with a unilateral paired 2-sample Student's t test). It can be partially explained because the threshold used in the soft core method induces a 'stair-step effect' along the rarefaction curves depending on the number of genomes sampled. This is illustrated on S2 Fig showing a step every 20 genomes (i.e. corresponding to $20 = \frac{100}{100-95}$ where 95% is the threshold of presence used) on the soft core curve of *L. plantarum*. We found a total of 20 species having atypical values of $\gamma$-tendency (absolute value above 0.05) and/or IQR area (above 15 000) for the soft core and only 2 species for the persistent genome of PPanGGOLiN, which are *Bacillus anthracis* and *Burkholderia cenocepacia*. For *B. cenocepacia*, it could be explained by the high heterogeneity of its shell (see next section), which is made of several partitions and complicates its distinction from the persistent genome during the process of partitioning. For *Bacillus anthracis*, the source of variability to define the persistent genome is a result of an incorrect taxonomic assignation in GenBank of about 17% of the genomes that are, according to the Genome Taxonomy DataBase (GTDB) [36], actually *B. cereus* or *B. thuringiensis*. This issue was not detected by our taxonomy control procedure because these species are at the boundary of the conspecific genomic distance threshold used (see Materials and methods). Some of persistent gene families of *bona fide B. anthracis* may therefore shift between persistent or shell partitions depending on the resampling. Excluding these misclassified genomes, we predicted a larger persistent genome than the one of the initial full set of genomes (about a thousand gene families more) with a $\gamma$-tendency much closer to 0 (-0.017 versus a $\gamma$-tendency of 0.036 for the soft core genome) and a lower IQR area (8367.0 vs 32167.1). Altogether, these results suggest that our approach provides a more robust partitioning of gene families in the

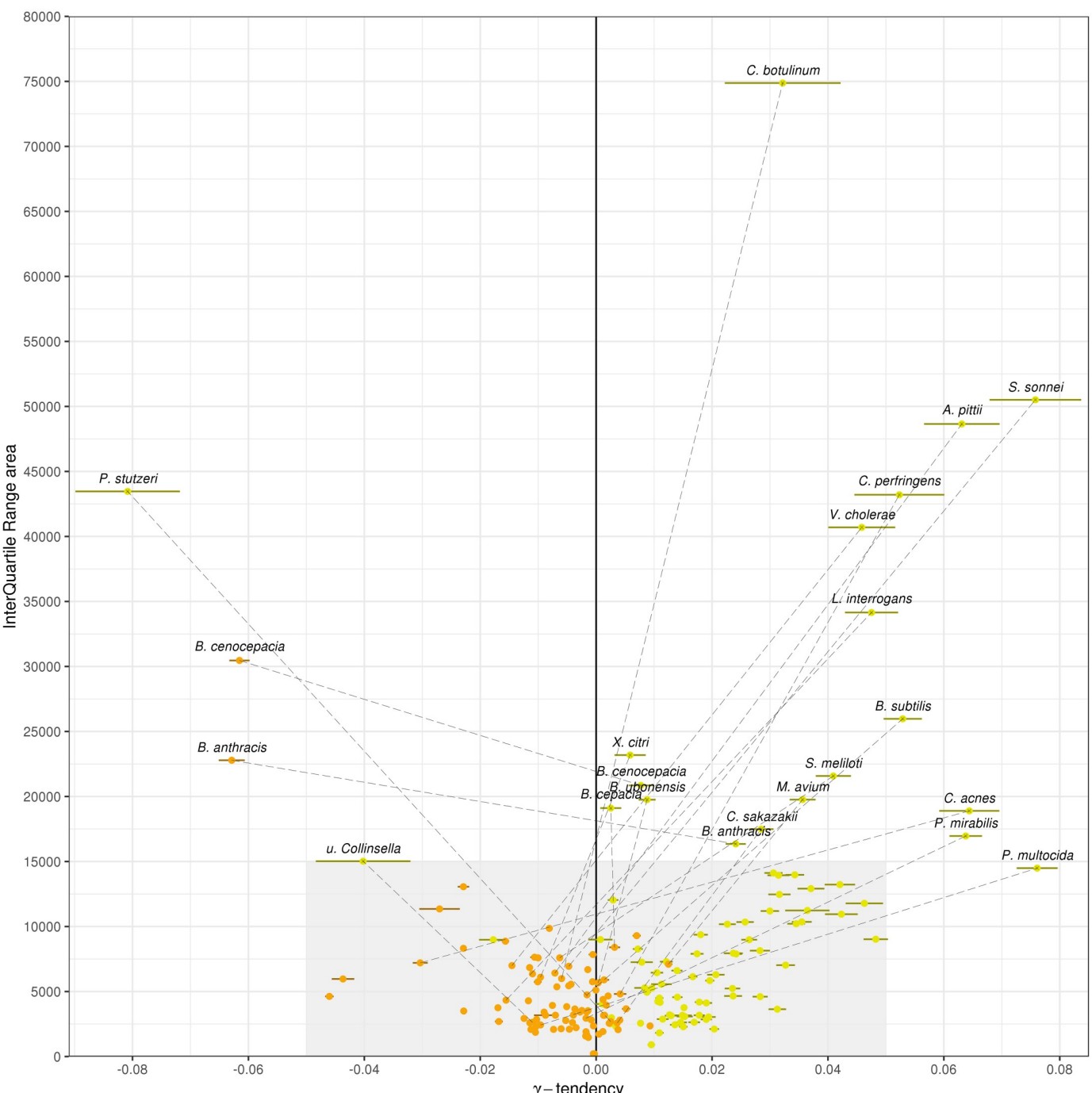

**Fig 4. γ-tendencies and IQR areas of the persistent and the soft core rarefaction curves.** Each of the 88 most abundant species in GenBank are represented by two points: orange points correspond to the PPanGGOLiN persistent values and yellow points to the ones of the soft core (≥95% of presence). A dashed line connects the 2 points if either the soft core or the persistent values are not in the range of the grey area ($-0.05 \leqslant \gamma \leqslant 0.05$ and $0 \leqslant IQR_{\text{area}} \leqslant 15000$). The colored horizontal bars show the standard errors of the fitting of rarefaction curves via the Heaps' law.

persistent genome than the use of arbitrary thresholds. Indeed, the statistical method behind PPanGGOLiN uses directly the information of the gene family P/A whereas the soft core is based only on frequency values. PPanGGOLiN can then classify families with similar frequencies in different partitions by distinguishing them according to their pattern of P/A in the

matrix and their genomic neighborhood. The main drawback of using family frequency to partition pangenomes is that even if it was possible to determine the best threshold for each species it would still not take into account that some persistent gene families may have atypically low frequency. This may be due to high gene losses in the population or technical reasons like belonging to a genomic region that is difficult to assemble (i.e. genes that are missing or fragmented in draft genome assemblies).

## Shell structure and dynamics

Two types of pangenome evolution dynamics are generally distinguished: open pangenomes and closed ones [1, 2, 35]. From rarefaction curves, the dynamics of pangenomes can be assessed using the $\gamma$-tendency of a Heaps' law fitting (see Materials and methods). A low $\gamma$-tendency means a rather closed pangenome whereas a higher $\gamma$-tendency means a rather open pangenome. A closed pangenome rigorously means a stabilized pangenome and we found no species obeying this strict criterion (that is to say $\gamma = 0$). This suggests that instead of using binary classifications for pangenomes, it is more useful to quantify the degree of openness of pangenomes given the flux of horizontal gene transfers and gene loss [7]. We computed rarefaction curves for the 88 studied species and determined the $\gamma$-tendency for different pangenome components (see S1 File and S3 Fig). The distribution of $\gamma$ values of the PPanGGOLiN shell genome shows a greater amplitude of values than the other components of the pangenome such as the whole pangenome or the accessory component. This indicates that the main differences in terms of genome dynamics between species seem to reside in the shell genome.

As expected, we found a positive correlation (Spearman's $\rho$ = 0.46, P = 8.2e-06) between the total number of shell gene families in a species and the $\gamma$-tendency of the shell (S4 Fig). This means that species with high $\gamma$-tendency do accumulate genes that are maintained and exchanged in the population at relatively low frequencies, suggesting they may be locally adaptive. More surprisingly, although one could expect that larger genomes have a larger fraction of variable gene repertoires, the fraction of shell and cloud genes per genome does not correlate with the genome size (Spearman's $\rho$ = 0.007, P = 0.95, Fig 5). The results remain qualitatively similar when analyzing the shell or the cloud separately (see S5 and S6 Figs). During this analysis, we noticed that, among host-associated bacteria with relatively small genomes (between $\approx$2000 and $\approx$3000 genes), three species (*Bifidobacterium longum*, *Enterococcus faecium* and *Streptococcus suis*) have a high fraction of shell genes ($>$ 28%) but low shell $\gamma$-tendency. Two of them (*B. longum* and *E. faecium*) are found in the gut of mammals and the third (*S. suis*) in the upper respiratory tract of pigs. They differ from other host-associated species in our dataset that are mainly human pathogens (e.g. bacteria of the genus *Corynebacterium*, *Neisseria*, *Streptococcus*, *Staphylococcus*) and have a low fraction of shell genes ($<$ 20%). It is possible that these three species have specialized in their ecological niches while maintaining a large and stable pool of shell genes for their adaptation to environmental stress. Further analysis would be required to confirm this hypothesis.

We then investigated the importance of the phylogeny of the species on the patterns of P/A of the shell gene families (shell structure). To this end, Spearman's rank correlations were computed between a Jaccard distance matrix generated on the basis of patterns of P/A of the shell gene families and a genomic distance obtained by Mash pairwise comparisons between genomes [37]. Mash distances were shown to be a good estimate of evolutionary distances for closely related genomes [38]. This correlation was examined in relation to the fraction of gene families that are part of the shell genome for each species (Fig 6). We observed that species with a high fraction of shell ($>$ 20% of their genome) have a shell structure that is mainly explained by the species phylogeny (i.e. shell P/A are highly correlated with genomic distances,

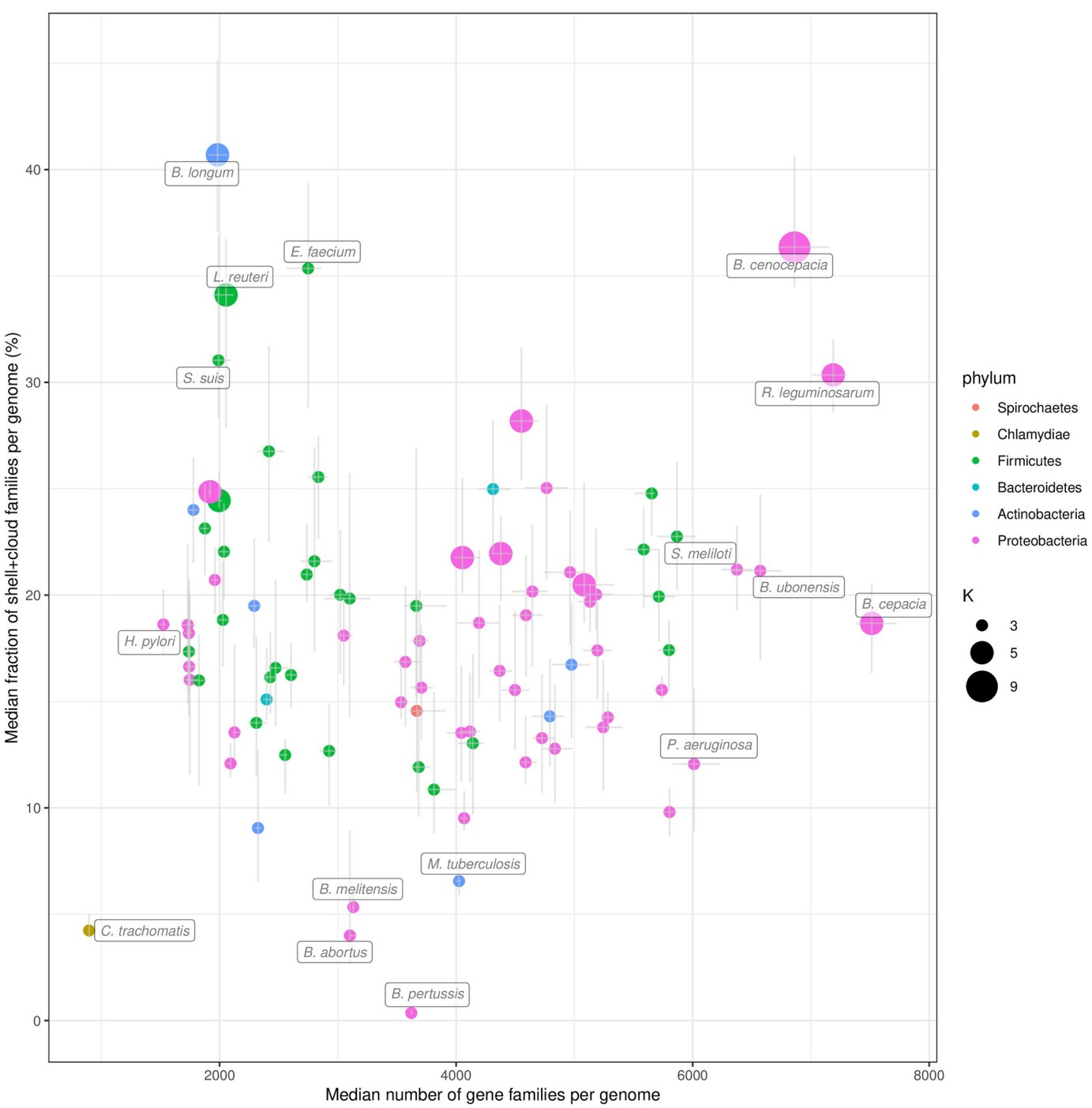

**Fig 5. Fraction of the variable (shell + cloud) families per genome compared to the number of gene families.** The results for the 88 most abundant species in GenBank are represented. The error bars show the interquartile ranges of the two variables. The points are colored by phylum and their size corresponds to the number of partitions (K) used.

Spearman's $\rho > 0.75$). In addition, PPanGGOLiN predicts a number of partitions (K) for these species often greater than 3. Hence, their shell is more heterogeneous between subclades and becomes structured in several partitions whereas for species with a single shell partition the shell is less structured, possibly indicating many gene exchanges between strains from different

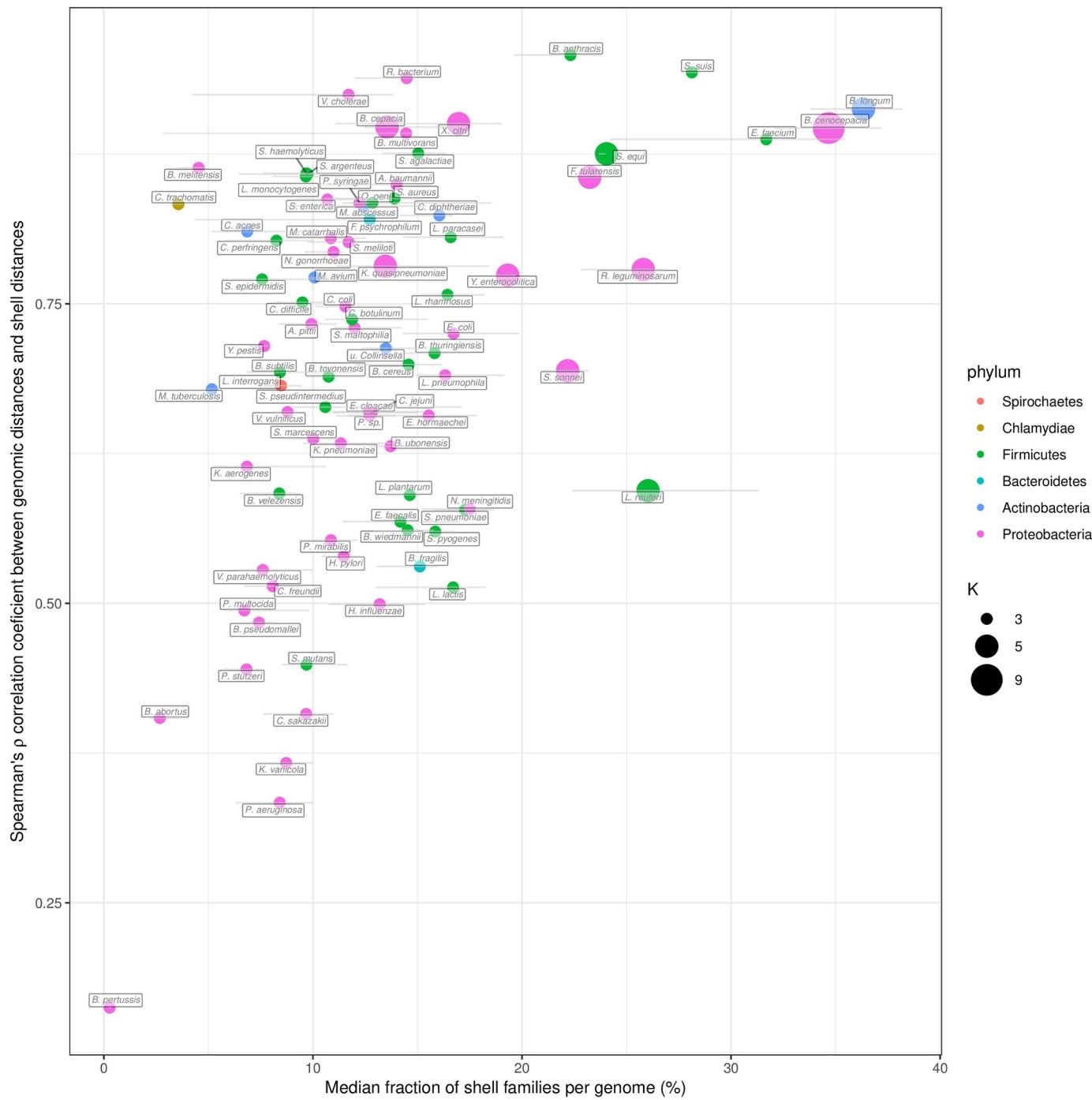

**Fig 6. Spearman's $\rho$ correlation coefficients between the shell genome presence/absence patterns and the MASH genomic distances compared with the shell fraction per genome.** The results for the 88 most abundant species in GenBank are represented. The error bars show the interquartile ranges of the shell fraction. The points are colored by phylum and their size corresponds to the number of partitions ($K$) used.

lineages. Among the nine species with a large shell genome (excluding *B. anthracis* due to taxonomic assignation errors), only two of them (*Shigella sonnei* and *Lactobacillus reuteri*) showed a relatively low correlation of their shell structure with the phylogeny (Fig 6). For *S. sonnei*, this could be explained by a high number of gene losses in the shell of this species that result

from convergent gene loss mediated by insertion sequences (preprint: [39]). For *L. reuteri*, these bacteria colonize the gastrointestinal tract of a wide variety of vertebrate species and have diversified into distinct phylogenetic clades that reflect the host where the strains were isolated, but not their geographical provenance [40]. As illustrated in S7 Fig, the shell of *L. reuteri* shows patterns of P/A that are only partially explained by the species phylogeny. Indeed, we observed clusters of families present across strains from distinct lineages that could contain factors for adaptation to the same host. In contrast, the shell structure of *B. longum* strongly depends on phylogenetic distances showing a clear delineation of adult and infant strains that have specialized into two subspecies (see S8 Fig).

We would like to stress the importance of the shell in the study of the evolutionary dynamics of bacteria. The shell content reflects the adaptive capacities of species through the acquisition of new genes that are maintained in the population. We found that the proportion of shell genes does not increase with the genome size. Instead, the shell accounts for a large fraction of the genomes of species when it is structured in several partitions. We can assume that those species are made of non-homogeneous subclades harboring specific shell genes which contribute to the specialization of the latter. Finally, it could be of interest to associate phenotypes to patterns of shell families that co-occur in different lineages independently of the phylogeny.

## Analysis of Metagenome-Assembled Genomes in comparison with isolate genomes

The graph approach should make our tool robust to gaps in genome data, making it a useful tool to analyze pangenomes obtained from MAGs. To test this hypothesis, we built the pangenomes of the Species-level Genome Bins (SGBs, clusters of MAGs that span a 5% genetic diversity and are assumed to belong to the same species) from the recent paper of Pasolli *et al.* [41]. This study agglomerated and consistently built 4 930 SGBs (154 723 MAGs) from 13 studies focussed on the composition of the human microbiome. We skipped the quality control step (already performed by the authors), and computed the pangenomes following the procedure we used for the GenBank species. The only parameter which differs is the *K* value which is set to 3 as the detection of several shell partitions is difficult for MAGs because of their incompleteness. To make the comparison with GenBank species, SGBs were grouped according to their estimated species taxonomy (provided by the supplementary table S4 of [41]). In this table, we noticed potential errors in the taxonomic assignation of two species (*Blautia obeum* and *Chlamydia trachomatis* corresponding to SGBs 4844 and 6877, respectively) and thus excluded them from the analysis. Keeping the same constraint as previously, only species with at least 15 genomes in both MAGs and GenBank were used for the comparison. A total of just 78 species (corresponding to 151 SGBs) could be analyzed as a lot of microbiome species are laborious to cultivate and thus less represented in databanks (see S2 File). Then, we compared the MAG pangenome partitions predicted by PPanGGOLiN with those obtained with GenBank genomes. To perform this, we aligned MAG and GenBank families for each species and computed the percentage of common families for each partition (see Materials and methods for details and S2 File for detailed results).

We observed that the size of the estimated persistent genome of MAGs is similar to the one of GenBank genomes for most species (Fig 7). In 55 out of the 78 species, the absolute fold change of persistent size is less than 1.2 and 90% (SD = 5%) of its content is common between MAGs and GenBank genomes. The 23 other species with more important differences showed smaller persistent genomes with only 60% (SD = 15%) of the persistent genome of GenBank being found in MAGs. For these species, the PPanGGOLiN method missed a fraction of the persistent genome due to the incompleteness of MAGs. Indeed, in such cases, the

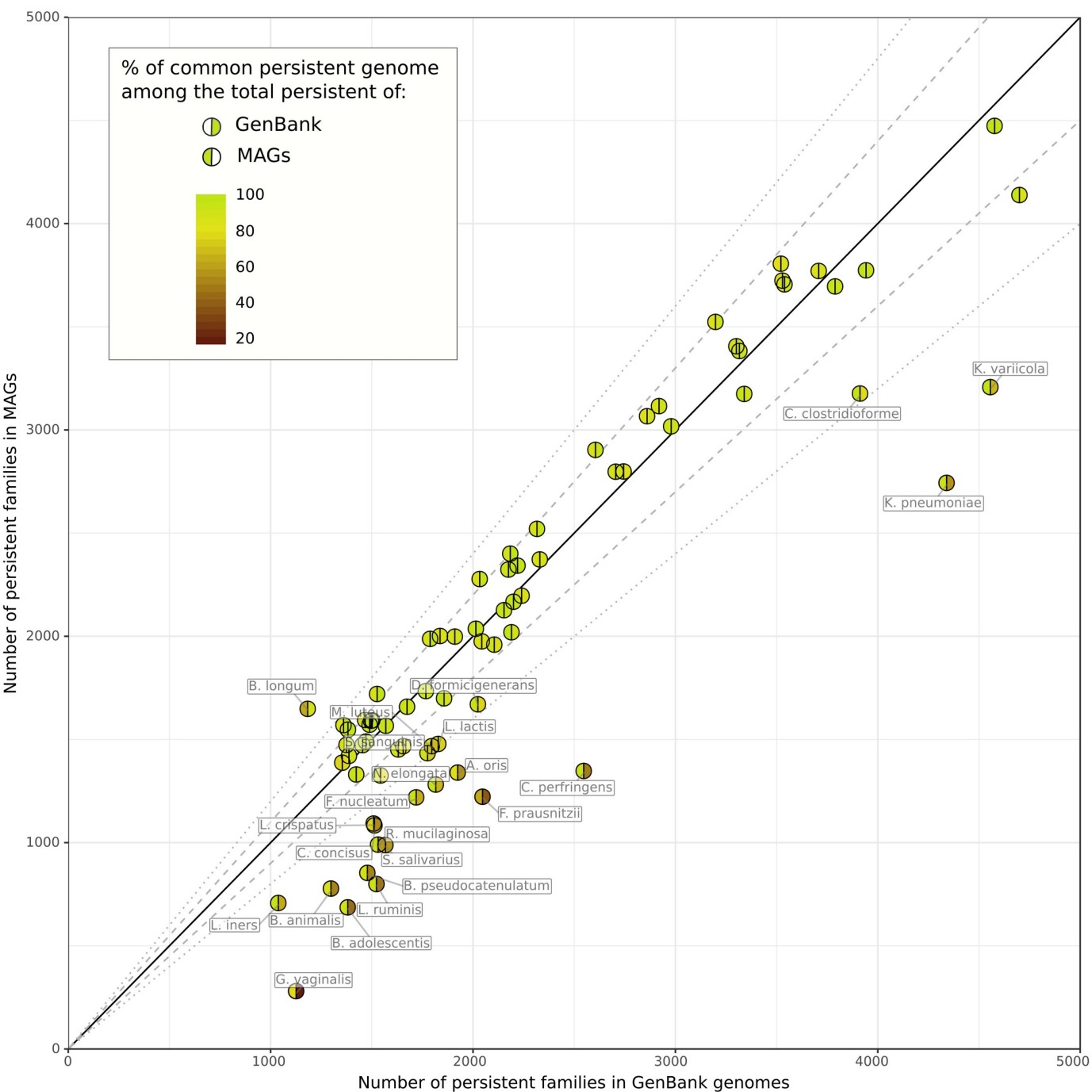

**Fig 7. Illustration of the persistent genome overlaps between GenBank genomes and MAGs.** Results for 78 species are represented. The colors of the hemispheres provide the percentage of common persistent gene families among the total persistent of MAGs (left hemisphere) or GenBank genomes (right hemisphere). The solid, dashed and dotted lines indicate the identity, a fold change of 1.1 and a fold change of 1.2 between the persistent genome sizes.

missing gene families are mostly classified in the shell of the MAGs which contains 32% (SD = 11%) of the GenBank persistent families. Nevertheless, 89% (SD = 9%) of the MAG persistent families match the GenBank ones, meaning that PPanGGOLiN correctly assigned persistent families for MAGs even if the persistent genome of these 23 species is incomplete.

However, two species, *Bifidobacterium longum* and *Faecalibacterium prausnitzii*, have less than 75% of their MAG persistent families in common with GenBank ones. For *B. longum*, this could be explained by the fact that the MAGs were obtained mostly from human adult samples while this species in databanks are from a broader host range (infants and pigs). It means that the MAG persistent might contain additional genes related to host-specificity. As a matter of fact, 412 gene families from the MAG persistent (25% of the total MAG persistent) are found in the GenBank shell which supports our hypothesis. For *F. prausnitzii*, the differences might be explained by a poor estimation of the persistent using GenBank data due to the low number of considered genomes (17 genomes versus 4232 MAGs). As expected, the soft core (based on the usual threshold of 95% presence) is unrealistically low in the MAG species with only ≈98 gene families on average and only 4 species out of 78 having more than 500 families classified in the soft core (see S2 File). Hence, the soft core approach is not well adapted to the analysis of MAGs. Furthermore, using lower thresholds of presence is not adequate because defining a unique threshold for all the families misses the heterogeneity of gene family presence in MAGs.

To explore the diversity within the pangenome of each species, we compared the shell of GenBank genomes and MAGs for the 55 ones with similar persistent genomes. Interestingly, we observed for all the 55 species only a partial overlap between the MAGs and GenBank shells (see S9 Fig). Indeed, as the MAGs are obtained only from a specific environment (i.e. the human microbiome), the diversity of GenBank is not fully captured by MAGs. It is especially the case for most of the Firmicutes and Proteobacteria. Conversely, most of the MAGs of Bacteroidetes phylum cover more than half of GenBank diversity while containing a large fraction of shell genes that are lacking in the shell of isolate genomes (i.e. less than 45% of the families are represented in the shell of GenBank). As already reported by Pasolli *et al.* [41], this confirms that the MAGs considerably improve the estimate of the genetic diversity of Bacteroidetes which are key players in the gut microbiome.

In summary, we have shown that PPanGGOLiN is able to provide an estimation of the persistent genome even using MAGs, which may miss significant numbers of genes and be contaminated by fragments from other genomes. This is especially the case for the accessory genome because its assembly coverage and nucleotide composition generally differ from those of the persistent genome making the binning of these regions more difficult. Nevertheless, PPanGGOLiN is able to find shell gene families in MAGs bringing new genes that may be important for species adaptation in the microbiome. Hence, it enables further analyses, even for uncultured species lacking reference genomes, such as the reconstruction of the core metabolism from the persistent genome to predict culture media or the study of the landscape of horizontally transferred genes within species.

## Conclusion

We have presented here the PPanGGOLiN method that enables the partitioning of pangenomes in persistent, shell and cloud genomes using a gene family graph approach. This compact structure is useful to depict the overall genomic diversity of thousands of strains highlighting variable paths made of shell and cloud genes within the persistent backbone. The statistical model behind PPanGGOLiN makes a more robust estimation of the persistent genome in comparison to classical approaches based on gene family frequencies in isolate genomes and also in MAGs. The definition of shell partitions based on statistical criteria allowed us to understand genome dynamics within species. We observed different patterns of shell with regard to phylogeny that may suggest different adaptive paths for the diversification of the species. It should be stressed that genome sampling is one of the main limitations of pangenome

studies and can therefore influence PPanGGOLiN partitioning especially for the shell genome. An improvement in the method could be to normalize the data to remove sampling bias. But as suggested by Brockhurst *et al.* [42], this issue should first be examined from a biological perspective by collecting and analyzing genomes from ecologically coherent microbial populations or ecotypes.

Future applications of PPanGGOLiN could include the prediction of genomics islands within the shell and cloud genomes. A first version of this application (Bazin *et al.*, in preparation) is already integrated in the MicroScope genome analysis platform [43]. Next, it would be interesting to determine the architecture of these variable regions by predicting conserved gene modules using information on the occurrence of families and their genomic neighborhood in the pangenome graph. Regarding metagenomics, pangenome graphs of PPanGGOLiN could be used as a reference (i.e. instead of individual genomes) for species quantification by mapping short or long reads on the graph to compute the coverage of the persistent genome. Indeed, each gene families of the partitioned pangenome graph could embed a variation graph as an alignment template [19]. Moreover, coverage variation in the shell or cloud genomes could allow the detection of strain-specific paths in the graph that are signatures of distinctive traits within microbiotes.

To conclude, the graph-based approach proposed by PPanGGOLiN provides an effective basis for very large scale comparative genomics and we hope that drawing genomes on rails like a subway map may help biologists navigate the great diversity of microbial life.

## Materials and methods

To explain the partitioning of pangenomes, we first need to describe the method based on the P/A matrix only (BinEM) and then the method built upon it that uses the pangenome graph to improve the partitioning (NEM).

### Modeling the P/A matrix via a multivariate Bernoulli Mixture Model

PPanGGOLiN aims to classify patterns of P/A of gene families into $K$ partitions ($K \in \mathbb{N}$; $K \geqslant 3$). Input data consists of a binary matrix $X$ in which a $x_{ij}$ entry is 1 if family $i$ is present in a genome $j$ and 0 otherwise (Fig 1) where $1 \leqslant i \leqslant F$ in each of the $F$ gene families and $1 \leqslant j \leqslant N$ in each of the $N$ genomes. A first approach for partitioning the data relies on a multivariate Bernoulli Mixture Model (BMM) estimated through the Expectation-Maximization (EM) algorithm [44] (named the BinEM method). The number of partitions $K$ may be greater than 3 (persistent, shell and cloud) due to the possible presence of antagonist P/A patterns among the different strains of a species. Therefore, two of the partitions will correspond to the persistent and cloud genome and a number of $K - 2$ partitions will correspond to the shell genome. The value of $K$ can be either provided by the user or determined automatically (see next section).

In the BMM, the matrix comprises data vectors $X_i = (x_{ij})_{1 \leqslant j \leqslant N}$ describing P/A of families, which are assumed to be independent and identically distributed with a mixture distribution given by:

$$P(X_i = (x_{ij})_{1 \leqslant j \leqslant N}) = \sum_{k=1}^{K} \pi_k \prod_{j=1}^{N} \epsilon_{kj}^{|x_{ij} - \mu_{kj}|} (1 - \epsilon_{kj})^{1 - |x_{ij} - \mu_{kj}|}$$

where $\pi = (\pi_1, \ldots, \pi_k, \ldots, \pi_K)$ denotes the mixing proportions satisfying $\pi_k \in [0, 1]$; $(\sum_{k=1}^{K} \pi_k) = 1$ and where $\pi_k$ is the unknown proportion of gene families belonging to the k$^{th}$ partition. Moreover, $\mu_k = (\mu_{kj})_{1 \leqslant j \leqslant N} \in \{0; 1\}^N$ are the centroid vectors of P/A of the k$^{th}$ partition representing the most probable binary states and $\epsilon_k = (\epsilon_{kj})_{1 \leqslant j \leqslant N} \in [0, \frac{1}{2}]^N$ are the unknown

vectors of dispersion around $\mu_k$. The default values of the dispersion vector $\epsilon_k$ associated to each centroid vector $\mu_k$ are constrained to be identical for all the $\epsilon_{kj}$ of a specific $k$ partition (for all the genomes of a specific partition) in order to avoid over-fitting but it is possible to release this constraint. The parameters of this model, as well as corresponding partitions, are estimated by the EM algorithm. To speed up the computation of the EM algorithm, a heuristic is used to initialize the BMM parameters in order to converge to a relevant partitioning using fewer EM-steps. This heuristic consists in setting $\pi_k$ with equiprobable proportions equal to $1/K$ while the $\epsilon_{kj}$ and $\mu_{kj}$ parameters are initialized triangularly.

Given $s = 1/\lceil K/2 \rceil$, the triangular initialization consists of:

$$\{\mu_{kj}\}_{1 \leqslant k \leqslant K/2, 1 \leqslant j \leqslant N} = 1$$

$$\{\mu_{kj}\}_{K/2 < k \leqslant K, 1 \leqslant j \leqslant N} = 0$$

$$\{\epsilon_{kj}\}_{1 \leqslant k \leqslant K/2, 1 \leqslant j \leqslant N} = s \cdot k$$

$$\{\epsilon_{kj}\}_{K/2 < k \leqslant K, 1 \leqslant j \leqslant N} = s \cdot (K - k + 1)$$

An interesting consequence of this initialization is that the persistent genome will be the first partition ($k = 1$) while the cloud genome will correspond to the last partition ($k = K$). This particular initialization solves the classical label switching problem in our context.

## Partitioning of the P/A matrix

To perform the partitioning of the P/A matrix, each gene family $i$ must be allocated to a single partition. The variables $\{Z_i\}_{1 \leqslant i \leqslant F}$ with a state space $\{1, \ldots, K\}$ indicate the partition to which each gene family $i$ belongs. Therefore, once the NEM parameters are optimized, the method automatically assigns the gene families to their most probable partition $z_i$ according to the model if their estimated posterior probability is above 0.5. If no partition can be assigned in this way, then the gene family is assigned to the shell (partition with intermediate frequency).

## Selection of the optimal number of partitions ($K$)

To determine the optimal $K$, named $\hat{K}$, the algorithm runs multiple partitionings with increasing values of $K$. After a few steps of the EM algorithm (10 steps by default), the Integrated Completed Likelihood (*ICL*) [45] is computed for each $K$. The *ICL* corresponds to the Bayesian Information Criterion (*BIC*) [46] penalized by the estimated mean entropy and is calculated as:

$$ICL(K) = BIC(K) - \sum_{k=1}^{K}\sum_{i=1}^{F} p(z_i \mid X, \hat{\theta}, k) \log\left(p(z_i \mid X, \hat{\theta}, k)\right); \forall p(z_i \mid X, \hat{\theta}, k) > 0$$

and

$$BIC(K) = \log \mathbb{P}_K(X \mid \hat{\theta}) - 1/2 \, dim(K) \log F$$

where $\log \mathbb{P}_K(X \mid \theta)$ is the data log-likelihood under a multivariate BMM with $K$ partitions and $\theta = (\{\pi_k\}_{1 \leqslant k \leqslant K}, \{\mu_{kj}\}_{1 \leqslant k \leqslant K, 1 \leqslant j \leqslant N}, \{\epsilon_{kj}\}_{1 \leqslant k \leqslant K, 1 \leqslant j \leqslant N})$. This log-likelihood can be calculated as follows:

$$\log \mathbb{P}_K(X \mid \theta) = \sum_{i=1}^{F} \log\left(\sum_{k=1}^{K} \pi_q \prod_{j=1}^{N} \epsilon_{kj}^{|x_{ij} - \mu_{kj}|} \left(1 - \epsilon_{kj}\right)^{1 - |x_{ij} - \mu_{kj}|}\right)$$

Moreover, $\hat{\theta}$ is the maximum likelihood estimator (approximated through the BinEM

algorithm) and $dim(K)$ is the dimension of the parameter space for this model. Here, $dim(K) = K(N + 2)$ if the dispersion vector $\epsilon_k$ associated to each centroid vector $\mu_k$ is constrained to be identical for all the $\epsilon_{kj}$ of a specific $k$ partition and $dim(K) = K(2N + 1)$ if the dispersion vector $\epsilon_k$ is free. Relying on this criterion, the best number of partitions is selected as $\hat{K} = \arg\min_{K} ((1 - \delta_{ICL})ICL(K))$ where $\delta_{ICL}$ is a sufficiently small margin to avoid choosing a too high $K$ value that would provide no significant gain compared to a lower value of $K$ (by default $\delta_{ICL} = 0.05 \times (max(ICL) - min(ICL))$).

## Generation of the pangenome graph

PPanGGOLiN uses a graph-based representation to store and visualize pangenomes. In this graph, the nodes correspond to gene families and the edges to genetic contiguity (i.e. genes that are direct neighbors in a genome). Two nodes are connected if the corresponding gene families contain at least one pair of genes that are adjacent in a genome. Edges are labeled with the corresponding genome identifiers and weighted by the proportion of genomes sharing that link. This process results in a pangenome graph (see Fig 2 as an example).

Formally, a pangenome graph $G = (V, E)$ is a graph having a set of vertices $V = \{(v_i)_{(1 \leqslant i \leqslant F)}\}$ where $F$ is the number of gene families in the pangenome associated with a set of edges $E = \{e_{i \sim i'}\} = \{(v_i, v_{i'})\}$, $v_i \in V$, $v_{i'} \in V$ where the couple of vertices $(v_i, v_{i'})$ are gene families having their genes $(v_{i,j}, v_{i',j})$ adjacent on the genome $j$ and where the function $countNeighboringGenes$ $(v_i, v_{i'})$ counts the adjacency occurrences in the $N$ genomes. Each edge $\{e_{i \sim i'}\}$ has a weight $w_{i \sim i'}$ where $w_{i \sim i'} = \frac{1}{N}\sum_{j=1}^{N} countNeighboringGenes(v_{i,j}, v_{i',j})$.

## Partitioning via Neighboring Expectation-Maximization

From the graph previously described, the neighborhood information of the gene families is used to improve the partitioning results. Indeed, the BinEM approach described above is extended by combining the P/A matrix $X$ with the pangenome graph $G$. This relies on a hidden Markov Random Field (MRF) model whose graph structure is given by $G$. In this model, each node belongs to some unobserved (hidden) partitions which are distributed among gene families according to a MRF which favors two neighbors to be more likely classified in the same partition. Conditional on this hidden structure, the binary vectors of P/A are independent and follow a multivariate Bernoulli distribution with proportion vectors depending on the associated partition. This approach is called NEM, as it relies on the Neighboring Expectation-Maximization algorithm [47–49]. As such, NEM tends to smooth the partitioning by grouping gene families that have a weighted majority of neighbors belonging to the same partition. The previously introduced latent variables $\{Z_i\}_{1 \leqslant i \leqslant F}$, that indicate the partition to which each gene family belongs are now distributed according to a MRF. More precisely, they have the following Gibbs distribution:

$$\mathbb{P}(\{Z_i\}_{1 \leqslant i \leqslant F}) = W_\beta^{-1} \exp\left(\sum_{i=1}^{F}\sum_{k=1}^{K}\pi_k 1_{Z_i=k} + \beta \frac{F}{\sum_{i\sim i'}w_{i\sim i'}}\sum_{i\sim i'}w_{i\sim i'}1_{Z_i=Z_{i'}}\right)$$

where $1_A$ is the indicator function of event $A$ and the second sum concerns every pair $(i \sim i')$ of neighbor gene families. The parameter $\beta \geq 0$ corresponds to the coefficient of spatial regularity. The $\frac{F}{\sum_{i\sim i'}^{E}w_{i\sim i'}}$ is a corrector term ensuring that the strength of the spatial smoothing is balanced regardless of the number of gene families. Indeed, when the number of genomes ($N$) increases, the number of gene families ($F$) tends to be higher than the sum of the edge weights.

Finally,

$$W_\beta = \sum_{\{\tilde{z}_i\} \in \{1...K\}^F} \exp\left(\sum_{i=1}^{F}\sum_{k=1}^{K} \pi_k 1_{\tilde{z}_i = k} + \beta \frac{F}{\sum_{i \sim i'} w_{i \sim i'}} \sum_{i \sim i'} w_{i \sim i'} 1_{\tilde{z}_i = \tilde{z}_{i'}}\right)$$

is a normalizing constant. Note that $W_\beta$ cannot be computed, due to a large number of possible configurations. The degree of dependence between elements is controlled by the parameter $\beta$. Neighboring elements will be more inclined to belong to the same group with a higher value of this parameter. Here, the data vectors $(X_i)_{1 \leqslant i \leqslant F}$ are not independent anymore. However, conditional on the latent groups $(Z_i)_{1 \leqslant i \leqslant F}$, they are independent and follow the multivariate Bernoulli distribution:

$$\mathbb{P}(\{X_i\}_{1 \leqslant i \leqslant F} | \{Z_i\}_{1 \leqslant i \leqslant F}) = \prod_{i=1}^{F}\prod_{j=1}^{N} \epsilon_{Z_i,j}^{|x_{ij} - \mu_{Z_i,j}|} (1 - \epsilon_{Z_i,j})^{1 - |x_{ij} - \mu_{Z_i,j}|}.$$

Many different techniques may be used to approximate the maximum likelihood estimator in the hidden MRF. NEM relies on a mean-field approximation for the distribution of the latent random variables $Z_{i1 \leqslant i \leqslant F}$ conditional on the observations. It should be noted that the optimal number of partitions ($K$) is not determined automatically using NEM and is therefore first estimated using the BinEM approach.

## Issues resulting from high-dimensional statistics and parallelization

As plenty of statistical approaches, NEM is not adapted to high dimensional settings (i.e. whenever the condition $F >> N$ is not satisfied). This can occur in pangenomics as the discovery rate of new families in the pangenome slightly decreases when new genomes are added. Mathematical solutions to this problem seem to exist [50–52] for example via the weighting of genomes (based on their respective contribution to the pangenome diversity) or via sparse partitioning methods. An improvement of NEM should include these solutions and could be a perspective of this work.

Pangenome software must be designed to scale up to thousands of genomes. NEM scales quadratically with the number of genomes and is hard to parallelize. Thus, it leads to intensive computations when thousands of genomes are included in the analysis.

Our solution to the mentioned issues is to sample the genomes in chunks and to perform multiple partitioning in parallel. Each family must be involved in at least $N_{total}/N_{samples}$ samplings and will be partitioned only if it is classified in the same partition in at least 50% of the samplings where it is present (absolute majority). If some families do not respect this condition, we continue sampling until all gene families have been partitioned. Chunks have to be large enough to be representative, therefore a size of at least 500 genomes is advised.

## Analysis of isolate genomes and Metagenome-Assembled Genomes

To obtain the set of isolate genomes to be analyzed, we downloaded all archaeal and bacterial genomes (220 561 genomes) of the GenBank database at the date of the 17th of April 2019. We removed genome assemblies that do not respect quality control criteria defined by GenBank. They correspond to entries with an assembly status flag different from "status = latest" in the "assembly_status.txt" files. In addition, genomes were discarded if they had more than 1000 contigs or a L90 > 100. These filters allowed us to exclude poor quality assemblies, some of which may correspond to contaminated genomes and others to incomplete ones. For each species (identified by its NCBI species taxid), a pairwise genomic distance matrix was computed using Mash (version 2.0) [37]. To avoid redundancy, if several genomes are at a Mash

distance < 0.0001, only one was kept (the one having the lowest number of contigs). A single linkage clustering using SiLiX (version 1.2.11) [53] was then performed on the adjacency graph of the Mash distance matrix considering only distances below or equal to 0.06. This Mash distance corresponds to a 94% Average Nucleotide Identity (ANI) cutoff which is a usual value to define species [54]. Genomes that were not in the largest connected component were discarded to remove potential taxonomic assignation errors. Only species having at least 15 remaining genomes were then considered for the analysis. The list of all the GenBank assembly accessions used after filtering is available in S3 File. This dataset consists of 439 species encompassing 136 287 genomes (see S1 File). MAGs from the Pasolli *et al*. study [41] were downloaded from https://opendata.lifebit.ai/table/SGB. In this dataset, the genomes are already grouped in Species Genome Bins. These SGBs do not exactly match the GenBank taxonomy. Thus, SGBs assigned with the same species name (column "estimated taxonomy" in the supplementary table S4 of [41]) were merged to allow comparison with GenBank. SGBs that do not have a taxonomy assigned at the species level were not considered. A total of 583 species encompassing 698 SGBs and 71 766 MAGs were analyzed but only MAGs from 78 species were finally compared to GenBank genomes. To avoid introducing a bias in our analysis due to heterogeneous gene calling, GenBank annotations were not considered as they were obtained using a variety of annotation workflows. Genomes from GenBank and Pasolli *et al* were consistently annotated using the procedure implemented in PPanGGOLiN. Prodigal (version 2.6.2) [55] is used to detect the coding genes (CDS). tRNA and tmRNA genes are predicted using Aragorn (version 1.2.38) [56] whereas the rRNA are detected using Infernal (version 1.1.2) [57] with HMM models from Rfam [58]. In the case of overlaps between a RNA and a CDS, the overlapping CDS are discarded. Homologous gene families were determined using MMseqs2 (version 8-fac81) [59] with the following parameters: coverage = 80% with cov-mode = 0, minimal amino acid sequence identity = 80% and cluster-mode = 0 corresponding to the Greedy Set Cover clustering mode. PPanGGOLiN partitioning was executed on each species using the NEM approach with a parameter $\beta$ = 2.5. The nodes having a degree above 10 (which is the default parameter) were not considered to smooth the partitioning via the MRF. The number of partitions ($K$) was determined automatically for each NCBI species using a $\delta_{ICL}$ = 0.05 and iterating between 3 and 20 for the possible values of K. K was fixed at 3 for the MAG analysis. The partitioning was done using chunks of 500 genomes when there were more than 500 genomes in a species. To compare PPanGGOLiN results between MAGs and GenBank genomes for each species, the representative sequences of each MAG gene family (extracted using the mmseqs2 subcommand: "result2repseq") were aligned (using mmseqs2 "search") on those of GenBank genomes. If the best hit of the query had a sequence identity > 80% and a coverage > 80% of the target, the 2 corresponding gene families of each dataset were associated.

## Rarefaction curves

To represent the pangenome evolution according to the number of sequenced genomes, a multiple resampling approach was used. For each species with at least 100 genomes, 8 rarefaction curves showing the evolution of the pangenome and the persistent, shell, cloud, soft core, soft accessory, exact core and exact accessory components were computed for sample sizes of 1 to 100 genomes randomly drawn from the set of all genomes of the species. Each sample size was analyzed using 30 different samples. For each sample, the number of partitions $K$ is automatically determined between 3 and the $K$ obtained on all the genomes of the species. A nonlinear Least Squares Regression was performed to fit the rarefaction curves with Heaps' law $F = \kappa N^{\gamma}$ where $F$ is the number of gene families, $N$ the number of genomes, $\gamma$ the tendency of

the evolution and $\kappa$ a proportional factor [35]. Subset sizes $\leq$ 15 were not used for the fitting as they are sometimes too variable to ensure a good fitting. The function "scipy.optimize.curve_fit" of the Python scipy package (version 1.0.0), based on the Levenberg-Marquardt algorithm, was used to fit the rarefaction curves. For each subset size, the median and quartiles were calculated to obtain a ribbon of interquartile ranges (IQR) along the rarefaction curves. We call the area of this ribbon the IQR area (see S2 Fig as an example).

## PPanGGOLiN software implementation

PPanGGOLiN was designed to be a software suite performing the annotation of the genomic sequences, building the gene families and the pangenome graph before partitioning it. Users can also provide their own annotations (GFF3 or GBFF format) and gene families. The application stores its data in a compressed HDF5 file but can also return the graph in GEXF or JSON formats and the P/A matrix with the partitioning in CSV or Rtab files (similarly to the ones provided by Roary [34]). It also generates several illustrative figures, some of which are presented in the article. PPanGGOLiN was developed in the Python 3 and C languages and is intended to be easily installable on Linux and Mac OS systems via a BioConda package [60] (see https://bioconda.github.io/recipes/ppanggolin/README.html). The code is also freely available on the GitHub website at the following address: https://github.com/labgem/PPanGGOLiN.

## Supporting information

**S1 Fig. Density distributions of the gene family frequencies of each partition.** Results for the 88 most abundant species in GenBank are represented in addition with a global distribution of the gene family frequencies from all the species. Density values of the cloud genome above 100 (y-axis) were trimmed for visualization purpose. The dashed yellow vertical bars indicate the threshold of frequency ($\geq$95%) used to delimit the soft core genome.
(PDF)

**S2 Fig. Evolution of the persistent, shell, soft core and exact core metrics of Lactobacillus plantarum compared to the number of genomes.** The rarefaction curves represent the evolution of the partition sizes as a function of an increasing number of genomes in random subsets of genomes. Plain lines connect the medians while colored areas represent the interquartile ranges. A regression curve (bold dashed line) is drawn fitting all the points of each partition by the Heaps' law ($F = \kappa N^{\gamma}$). The total area of the interquartile ranges (IQR) is indicated for each partition.
(TIF)

**S3 Fig. Density distributions of the Heaps' law $\gamma$-tendencies.** These $\gamma$-tendencies were obtained by fitting a Heaps' law on rarefaction curves between subset sizes of 15 to 100 genomes in the 88 most abundant species in GenBank. The exact core median and exact accessory are not shown.
(TIF)

**S4 Fig. Shell $\gamma$-tendency compared to the total number of shell families normalized by the median number of gene families per genome in each species.** Results for the 88 most abundant species in GenBank are represented. The points are colored by phylum and their size corresponds to the number of partitions ($K$) used.
(TIF)

**S5 Fig. Fraction of shell families per genome compared to the number of gene families.**
Results for the 88 most abundant species in GenBank are represented. The points are colored
by phylum and their size corresponds to the number of partitions (*K*) used.
(TIF)

**S6 Fig. Fraction of cloud families per genome compared to the number of gene families.**
Results for the 88 most abundant species in GenBank are represented. The points are colored
by phylum and their size corresponds to the number of partitions (*K*) used.
(TIF)

**S7 Fig. Presence/Absence matrix of the shell genome of *L. reuteri* ordered by a Neighbor
Joining tree based on the MASH distances.** The leaves of the tree are colored by host or origin. This information was obtained from the metadata in GenBank files (host and isolation
source qualifiers).
(TIF)

**S8 Fig. Presence/Absence matrix of the shell genome of *B. longum* ordered by a Neighbor
Joining tree based on the MASH distances.** The leaves of the tree are colored by species clusters defined by the GTDB database (release R04-RS89), namely (*B. infantis* or *B. longum*).
"NA" values correspond to genomes not available in GTDB.
(TIF)

**S9 Fig. Illustration of the shell genome overlaps between MAGs or GenBank of 55 species.**
The x-axis represents the percentage of common shell of the GenBank shell while the y-axis
corresponds to the percentage of common shell of the MAGs shell. Diamonds and squares represent MAGs and GenBank genomes, respectively. They are colored by phylum and their size
indicates the number of genomes.
(TIF)

**S1 File. Table compiling all the metrics obtained from the pangenomes of the 439 GenBank species.** This is a CSV file.
(CSV)

**S2 File. Table compiling all the metrics obtained from the comparison of PPanGGOLiN
results between MAGs and GenBank genomes in 78 species.** This is a CSV file.
(CSV)

**S3 File. List of GenBank assembly accessions for the 439 studied species.** This is a TSV file
where each line corresponds to all the GenBank assembly accession used in this study for each
'species id' in the NCBI taxonomy.
(TSV)

## Acknowledgments

We acknowledge Alexandre Renaux and Jonathan Mercier for their preliminary insights on
pangenome graphs. We thank Mélanie Buy for drawing the PPanGGOLiN logo. Finally, we
thank Guilhem Royer, Valentin Sabatet, Johan Rollin, Mohammed-Amin Madoui, Tom Delmont, Nicolas Pons and Pierre Peterlongo for all their advice along this work.

## Author Contributions

**Conceptualization:** David Vallenet.

**Data curation:** Guillaume Gautreau, Adelme Bazin, Mathieu Gachet, Rémi Planel, Laura Burlot, Mathieu Dubois, Amandine Perrin.

**Formal analysis:** Guillaume Gautreau, Adelme Bazin.

**Investigation:** Guillaume Gautreau, Adelme Bazin.

**Methodology:** Guillaume Gautreau, Catherine Matias, Christophe Ambroise.

**Software:** Guillaume Gautreau, Adelme Bazin.

**Supervision:** David Vallenet.

**Visualization:** Guillaume Gautreau.

**Writing – original draft:** Guillaume Gautreau, Adelme Bazin, Eduardo P. C. Rocha, David Vallenet.

**Writing – review & editing:** Guillaume Gautreau, Adelme Bazin, Mathieu Dubois, Claudine Médigue, Alexandra Calteau, Stéphane Cruveiller, Catherine Matias, Christophe Ambroise, Eduardo P. C. Rocha, David Vallenet.

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
