## [Decision Letter · Decision Letter 0]

18 Dec 2019

Dear Dr Vallenet,

Thank you very much for submitting your manuscript, 'PPanGGOLiN: depicting microbial diversity via a partitioned pangenome graph', to PLOS Computational Biology. As with all papers submitted to the journal, yours was fully evaluated by the PLOS Computational Biology editorial team, and in this case, by independent peer reviewers. The reviewers appreciated the attention to an important topic but identified some aspects of the manuscript that should be improved.

We would therefore like to ask you to modify the manuscript according to the review recommendations before we can consider your manuscript for acceptance. Your revisions should address the specific points made by each reviewer and we encourage you to respond to particular issues Please note while forming your response, if your article is accepted, you may have the opportunity to make the peer review history publicly available. The record will include editor decision letters (with reviews) and your responses to reviewer comments. If eligible, we will contact you to opt in or out.raised.

- Supporting Information uploaded as separate files, titled 'Dataset', 'Figure', 'Table', 'Text', 'Protocol', 'Audio', or 'Video'.

We hope to receive your revised manuscript within the next 30 days. If you anticipate any delay in its return, we ask that you let us know the expected resubmission date by email at ploscompbiol@plos.org.

Sincerely,

Christos A. Ouzounis

Associate Editor

PLOS Computational Biology

William Noble

Deputy Editor

PLOS Computational Biology

[LINK]

Reviewer's Responses to Questions

**Comments to the Authors:**

Reviewer #1: "PPanGGOLiN: depicting microbial diversity via a partitioned pangenome graph", authored by G. Gautreau et. al. presents PPanGGoLiN; a graph-based method for the classification of gene families into persistent, cloud and one or several shell partitions, that can be used for prokaryotic genomes. The authors make a strong case on the relevance of the persistent genome instead of the more frequently used soft core genome. Moreover, they point out the value of the shell genome in the study of evolutionary dynamics. They also demonstrate the robustness of their method by applying it to metagenome-assembled genomes and comparing their results to the soft core metric.

The article is well written and the results are sufficiently supported by the text and figures. There are a few points, however, where I would have liked some more clarifications and which are described in detail below.

1-- For the illustration of the partitioned pangenome graph you analysed the A. baumannii pangenome (line 107). There you report the sizes of persistent, shell and cloud values derived from your method. How do you evaluate these numbers? Are they expected? How do they compare to existing literature (for example Chan 2015 which you mention in the introduction, or Hassan 2016)?

2-- Even though the principle of the PPanGGOLiN software is described in detail, the article does not provide much information about other existing software. It would be nice to have a few lines on the types of existing software (common features, novelties) and explain how PPanGGOLiN compares to them.

3-- When estimating the persistent genome in comparison to the soft core approach (line 173 and onwards) you are introducing the idea that some gene families can belong to the persistent group even if they are observed in surprisingly low frequencies (line 234). Can you give some more concrete metrics on which percentage of the persistent genome would belong to the soft core and which to the shell?

4-- My main concern with the article is regarding the interpretation of the idea of partitions. In lines 306-307 for example, the large number of partitions is assumed to be due to non-homogeneous subclades. Could variation in partitions also be influenced by genome sampling? And taking this thought one step further, have you tried comparing the partitions that are generated in case of genus or phylum pangenomes? How about in the same example but by gradually adding more genomes in the analysis?

5-- Some parts of the methods section do not have such a good use of the english language as the rest of the article. Below is a list of a few typos but it would be a good idea to scan the whole methods section for grammatical errors and typos.

line 287: due  due to

line 546: share together  shared

line 564: annotation  annotations

line 565: there are  they were

Reviewer #2: This paper describes a method for using graph structures to model bacterial pan-genomes, in terms of gene presence and absence. This approach is different from conventional approaches - it is aesthetically pleasing, and kind of makes intuitive sense. However, I'm not sure that finding different results with a different method necessarily means that this is a 'better' method. Yes, it's different - but benchmarking is difficult.

So I think that in a nutshell, the method is good, and should be published, but I do not think that the authors have convinced me that their method is necessarily 'better'.

The figures are good, but could be improved (especially the figure legends could be more clear) - Figure 1 is a nice overview, and the 'mobilome' in figure 2 is nicely visualized (although a black background is not optimal - white backgrounds can often give better visualizations). I like representing the species core-genomes on a single page, like Figure 3.

Lines 82-83 - the authors ASSUME that horizontally transferred genes ALWAYS go in to a few hot spots, and 'persistent genes' have conserved synteny (a word that is curiously missing from the entire manuscript!). Is it possible for a conserved gene to have different neighbors??

Lines 151-152 - this seems contradictory - if their method proposed can be used for low quality genomes, then why is it necessary to throw out the low-quality genomes?

Lines 190-191 - Yes, the 'most cited software' assumes 95% - but this does not necessarily make it correct or the 'best'. There are mathematical methods for testing this - see for example the "kneedle" method. Satopaa, V., Albrecht, J., Irwin, D., and Raghavan, B. (2011). Finding a “kneedle” in a haystack: Detecting knee points in system behavior. In 2011 31st International Conference on Distributed Computing Systems Workshops, pages 166–171.

**Have all data underlying the figures and results presented in the manuscript been provided?**

Reviewer #1: Yes

Reviewer #2: Yes

PLOS authors have the option to publish the peer review history of their article (what does this mean?). If published, this will include your full peer review and any attached files.

Reviewer #1: Yes: Anastasia Chasapi

Reviewer #2: Yes: David Ussery

---

## [Decision Letter · Decision Letter 1]

12 Feb 2020

Dear Dr. Vallenet,

We are pleased to inform you that your manuscript 'PPanGGOLiN: depicting microbial diversity via a partitioned pangenome graph' has been provisionally accepted for publication in PLOS Computational Biology.

Before your manuscript can be formally accepted you will need to complete some formatting changes, which you will receive in a follow up email. A member of our team will be in touch within two working days with a set of requests.

Best regards,

Christos A. Ouzounis

Associate Editor

PLOS Computational Biology

William Noble

Deputy Editor

PLOS Computational Biology

Reviewer's Responses to Questions

**Comments to the Authors:**

Reviewer #1: The authors have sufficiently answered to the points I raised and have accordingly modified their article. I, therefore, have no further comments and recommend the article for publication.

Reviewer #2: The authors have done a good job of addressing the reviewer's comments.

**Have all data underlying the figures and results presented in the manuscript been provided?**

Reviewer #1: Yes

Reviewer #2: Yes

PLOS authors have the option to publish the peer review history of their article (what does this mean?). If published, this will include your full peer review and any attached files.

Reviewer #1: Yes: Anastasia Chasapi

Reviewer #2: Yes: David Ussery

---

## [Editor Report · Acceptance letter]

10 Mar 2020

PCOMPBIOL-D-19-02015R1 

PPanGGOLiN: depicting microbial diversity via a partitioned pangenome graph

Dear Dr Vallenet,

I am pleased to inform you that your manuscript has been formally accepted for publication in PLOS Computational Biology. Your manuscript is now with our production department and you will be notified of the publication date in due course.

With kind regards,

Sarah Hammond
